# Mapping of shore area wetlands in Lake Tana Biosphere Reserve, Northwest Ethiopia using Sentinel-1A SAR and multi-source data

Yirga Kebede Wondim[ID][1]*, Ayalew Wondie Melese[1], Workiyie Worie Assefa[2]

1 Department of Aquatic and Wetland Ecosystem Management, School of Fisheries and Wildlife, College of Agriculture and Environmental Sciences, Bahir Dar University, Bahir Dar, Ethiopia, 2 Department of Biology and Blue Nile Water Institute, College of Science, Bahir Dar University, Bahir Dar, Ethiopia

* yirgukeba@gmail.com

## Abstract

Shore area wetlands (lacustrine fringe) play a critical role as ecotones that support biodiversity, provide habitats for spawning and refuge, and exhibit high levels of primary productivity. They facilitate significant exchanges of materials between aquatic and terrestrial ecosystems. To effectively manage and preserve these important resources, it is essential to understand their distribution, size, and dynamic changes. This study aimed to create an accurate map of shoreline wetlands using multi-temporal and multi-source data, wetland indicators such as wetland hydrology (WH), hydrophytic vegetation (HV), hydric soil (HS), and radar imagery from Sentinel-1A, employing Geomatica software. Additionally, ArcGIS software was used to map the topographic position (TP), Lake Bathymetry (LB), and HS indicators for wetlands. The analytical hierarchy process and weighted overlay methods were also applied in the mapping process for integrating all the indicators to obtain the final extent of shoreline wetlands. The TP wetland indicator map covered about 55,364 ha, while HS covered around 55,151 ha within a 3 km buffer from Lake Tana. The map of WH indicator for wetlands revealed that permanently inundated areas accounted for roughly 591,312 ha, and when temporarily inundated areas were included, the total coverage increased to 607,053 ha. HV, including invasive water hyacinth, covered over 74,772 ha. Overall, shoreline wetlands were predominantly located within three kilometers of the terrestrial area from Lake Tana, totaling 26,664 ha. The overall accuracy of land use and cover classification was recorded at 79%, with a Kappa statistic of 0.70, indicating that the resulting map is of acceptable quality. The integration of multi-temporal and multi-source data, along with wetland indicators and radar imagery from Sentinel-1A using Geomatica software, has provided valuable insights into the spatial distribution of shoreline wetlands in Lake Tana. The findings from this study will serve as an important reference for future research aimed at effectively managing and conserving these vital resources.

**Data availability statement:** All relevant data are within the manuscript and its supporting information files.

**Funding:** The author(s) received no specific funding for this work.

**Competing interests:** I have read the journal's policy and the authors of this manuscript have the following competing interests:The authors declare that they have no known competing financial interests or personal relationships that could have appeared to influence the work reported in this paper.

## Introduction

Shore area wetlands (lacustrine fringe) are important ecotones for biodiversity conservation, providing spawning and refuge habitats, exhibiting high primary productivity, and facilitating significant material exchange between aquatic and terrestrial ecosystems [1]. Despite their importance, lacustrine wetlands are among the most vulnerable ecosystems to human activities, particularly due to water level fluctuations [2]. Lake Tana, the largest freshwater lake in Ethiopia, is rich in lake-associated shoreline wetlands. These wetlands support spawning and juvenile habitats for several important fish species, including the unique endemic *Labeobarbus* [3], as well as a refuge for zooplankton [4]. Moreover, they serve as the feeding and breeding ground for various bird species such as the *common crane*, *wattled crane*, and *crowned crane* [5]. However, the wetlands of Lake Tana have faced severe threats from multiple anthropogenic activities; including recession agriculture, water level fluctuations, and siltation. The invasion of the lake by *Eichhornia crassipes* (water hyacinth) since 2011 has further exacerbated these challenges, as it quickly invades and dominates native wetland plant species [6].Thus, it is essential to understand the distribution, size, and dynamic changes of the shoreline wetlands to effectively manage and preserve these crucial resources.

The use of GIS and remote sensing technologies for mapping and monitoring wetland resources has been well-established [7,8]. Remote sensing technology has been utilized to map wetland areas or changes in land use/cover in wetlands during the past five decades [9–11]. In Egypt, remote sensing technology has been utilized to detect environmental changes and land surface temperature [12,13]. To create a thorough grasp of wetland dynamics throughout Africa, the work of [14] integrates carbon stock calculation, fragmentation analysis, and spatial mapping using GIS and remote sensing technologies. However, wetlands pose unique challenges for mapping due to their dynamic nature, environmental factors, and the limitations of traditional techniques [15]. Wetlands can be classified into various categories based on their hydrology, soils, water chemistry, vegetation, landscape position, and interests of the classifier. Most of the remote sensing techniques cannot detect water level changes beneath the vegetation in wetland ecosystems. Recently, the Synthetic Aperture Radar (SAR) technology has been applied effectively in conditions of persistent cloud, smoke, and haze, enabling the detection of water level changes across different wetland environments [16,17]. As an active microwave sensor, SAR can operate day or night, capturing images regardless of weather conditions [18]. This technology allows for the detection of water level changes under various weather situations by leveraging the 'double bounce' phenomenon [19,20].Despite these advantages, speckle noise in SAR images reduces the image's radiometric quality and complicates processing [21 ,22]. Fortunately, [23] demonstrated that utilizing an effective despeckling method can positively impact the precision of wetland classification.

The use of SAR data for wetlands studies has increased tremendously. Various sensors have been used in wetland investigations, including the ALOS Phased Array L-band Synthetic Aperture Radar (PALSAR), European Remote Sensing (ERS-1),

RadarSAT, ASAR, Japanese Earth Resources Satellite 1 (JERS-1), AIRSAR, and TerraSAR-X Sentinel-1 [24]. WH in areas such as the Everglades [19], Louisiana, and the Sian Ka'an in Yucatan [25] has been effectively assessed using Interferometric SAR (InSAR) observations. The same technique has been applied for multi-temporal monitoring of wetland water levels in the Florida Everglades [19], and has been combined with Radarsat-1 data to map fluctuations in water levels in coastal wetlands in southeastern Louisiana [18]. However, the use of radar data for the study of land use/cover, particularly wetlands, remains quite limited in Ethiopia. Only one study has been conducted on mapping the Dabus wetlands in Ethiopia by using Random Forest Classification of Landsat, PALSAR, and Topographic Data [11]. Additionally, a combination of sentinel-1 SAR and sentinel-2 MSI data has been employed for accurate urban land use-land cover classification in Gondar City [26].

Nevertheless, to our knowledge, there are currently no studies exploring the application of Sentinel-1A SAR for mapping shore area wetlands. Hence, this work aims to develop an accurate mapping of lacustrine wetlands using multi-temporal and multi-source data, indicators for wetlands, and radar imagery from Sentinel-1, all processed using Geomatica software. Specifically, this study will integrate spatial geo-information derived from topographic position and hydric soil indicators (essential for wetland identification) using ArcGIS software with data captured by the active sensor of Sentinel-1A SAR, which includes information on HV and WH. The specific research components of this study are as follows: (a) Mapping TP (including Digital Elevation Models (DEM) and its derivatives, as well as LB) and HS wetland indicators using various multi-source datasets and ArcGIS software; (b) Mapping HV and WH using Sentinel-1 SAR data by processing in the Geomatica Banff software package; (c) Finally, integrating all wetland indicators to map shore area wetlands in Lake Tana Biosphere Reserve. We expect that SAR remote sensing data alone could not produce satisfactory wetland classification accuracy; rather, mapping wetlands using multiple wetland indicator approach ensures a more accurate and comprehensive representation of wetlands.

## Materials and methods

### Description of the study area

Lake Tana (Fig 1) is the largest freshwater lake in Ethiopia. It is characterized by extensive shorelines and terrestrial wetlands [27], which provide essential spawning habitats for approximately half of the *Labeobarbu*s species, as well as three other commercially important fish species: Nile tilapia (*Oreochromis niloticus*), African catfish (*Clarias gariepinus*), and Beso (*Varicorhinus beso*) [3]. The juveniles of these fish species feed and grow in the lakeshore as well during the early years of their life [3], while the wetlands also serve as a refuge for zooplankton [4]. Additionally, these areas act as breeding grounds for various bird species, including the common crane (a wintering species), the endangered wattled crane, and the crowned crane [5].The lake and its associated wetlands provide immense socioeconomic benefits, supporting the livelihoods of thousands of people and contributing to economic activities through hydropower production and irrigation [28].Despite their importance, the wetlands, including Lake Tana, have faced challenges from both short- and long-term water level fluctuations and heavy sediment loads, threatening their sustainable use.

The shore area wetlands surround the lake (Fig 1) and are formed and maintained through lacustrine processes. At mean annual water depths of less than 2 meters, these wetlands experience either permanent or periodic flooding. Tana, an adjacent freshwater lake, supplies water to these wetlands. In addition to the lake water, precipitation, and surface inflows also contribute to these lacustrine fringe wetlands. During the rainy season, the lake and rivers, along with precipitation, serve as the primary water inputs to the wetlands [27].

Within the shore area wetlands of Lake Tana, including the Lake itself, several physical, chemical, and biological processes work to purify the inflowing water and buffer the lake against pollution [29] made the following generalizations after relating the circulation patterns of Lake Tana to the distribution of sediment, nutrients, and water hyacinth: (a) water, sediment, and associated nutrients, such as phosphorus generated from the eastern and northern catchments, had the longest retention time in the northeast of Lake Tana and as a result, most of the water is lost through evaporation while the

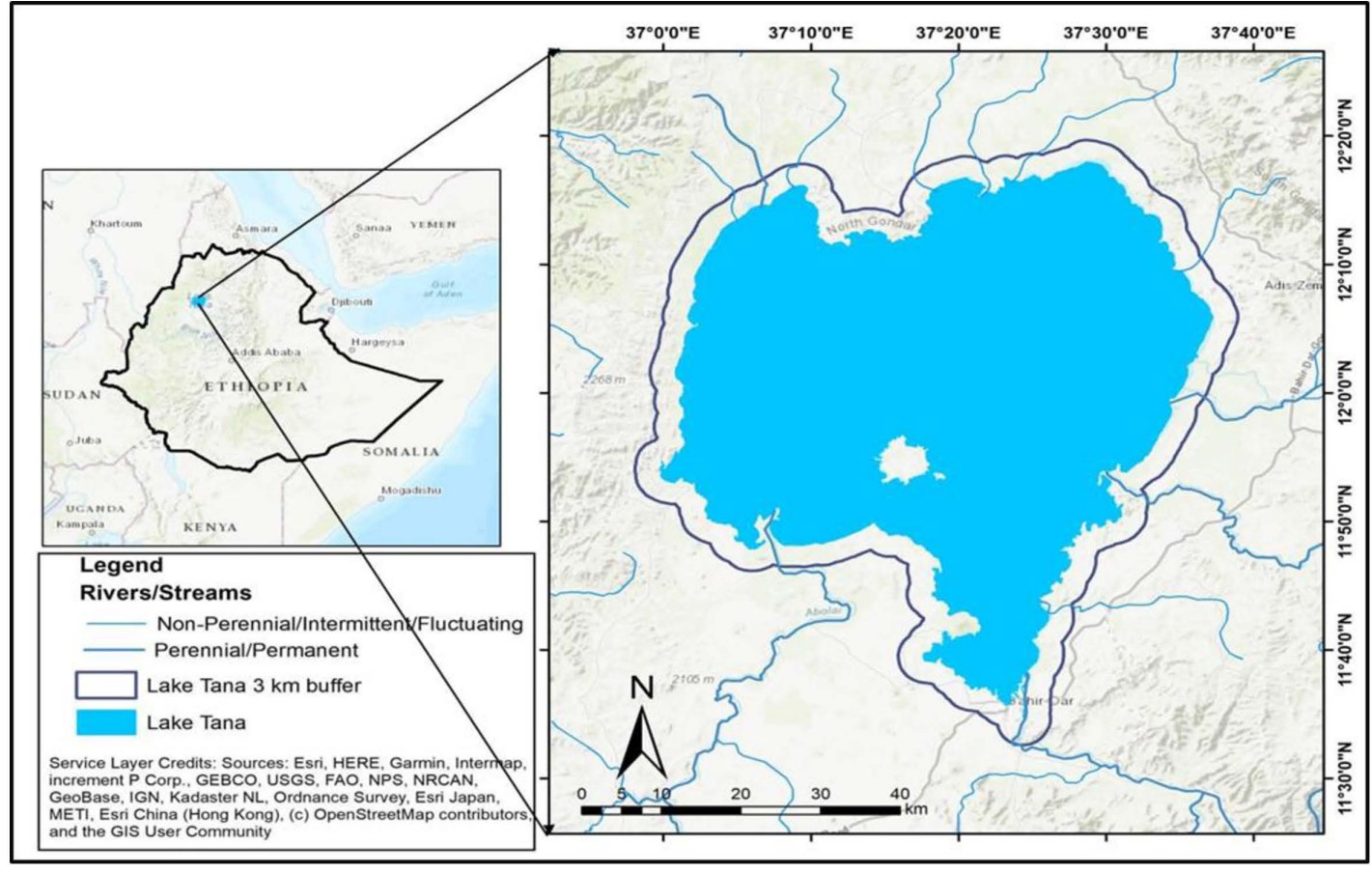

**Fig 1. Location map of the study area.** *Attribution: Esri, Delorme, HERE, Tom Tom, Intermap, Increment P Corp, GEBCO, USGS, FAO, NPS, NRCAN, GeoBase, IGN, Kadaster NL, Ordance Survey, Esri Japan, METI, Esri China(Hong Kong), swisstopo, MapmyIndia, and the GIS User Community.*

sediment and phosphorus are partly suspended in the area and absorbed in the lakebed in the northeast, and (b) sediments and nutrients delivered from Gilgel Abay, southwest of Lake Tana, flow out of the lake through the two outlets in a shorter retention time favored by the close locations with the outlets.

Wetlands rely on precipitation (rainfall), surface water, and groundwater for their hydrological needs. Because it serves as a major supply of water and affects water levels, flow patterns, and ultimately the ecology of the wetland, rainfall is essential to the development and survival of wetlands. The seasonal variation of rainfall in Lake Tana sub-basin is controlled by the northward and southward movement of the intertropical convergence zone and the mean annual rainfall varies from 1, 200–1, 600 mm [30]. Summer is the Lake Tana sub-basin's main rainy season, spanning from mid-June to mid – September, with the sub-basin's climate consisting mainly of tropical highland monsoon [31].Generally, the southern part of the Lake Tana sub-basin is wetter than the western and the northern parts.

## Mapping of shore area wetlands in Lake Tana

**Data types and sources.** In efforts to facilitate the mapping of shore area wetlands at Lake Tana, four key wetland indicators were selected: WH, HV, HS, and TP. These wetland indicators were extracted based on various data sources including field observations, secondary sources and remote sensing data (S1Table). Hence, in order to

supplement the wetland delineation and mapping in the study area wetland inventory, which comprises especially the hydrophytic vegetation survey, hydric soil field observation, wetland hydrology field observation, and elevation ground truth using a handheld GARMIN GPSmap 62s, it was conducted in the mapping procedure, which strengthens the mapping processes (remote sensing and secondary data sources). Using multiple seasons of data allows improved discrimination of wetland types by assessing the seasonal variations in phenology related to changes in plant structure. Therefore, in this study, wetland field inventory was conducted during May 2021 and October 2021 for wetland data. The bathymetry of Lake Tana and the shape file format of the Lake Tana sub-basin soil survey were from secondary data sources to delineate the boundaries of the wetlands from the lake-ward side and hydric soil indicator map, respectively.

A total of 10 Sentinel-1 GRD images in the Interferometric Wide swath (IW) mode (10 m resolution) with VV and VH polarizations from May 2021, the end of the peak dry season, and Nov 2021, the end of the peak wet season, were downloaded and processed to produce a hydrophytic vegetation indicator map and a wetland hydrology indicator map (S1Table).May marks the transition from the dry season to the onset of the wet season in our region. During this time, temporary wetlands and seasonal floodplains are typically at their minimum extent, and vegetation is less vigorous. SAR data from this period is instrumental in delineating permanent water bodies and areas of persistent inundation. October coincides with the peak of the wet season, when seasonal wetlands and floodplains reach their maximum extent. Elevated soil moisture and inundation levels, coupled with vigorous growth of hydrophytic vegetation, may result in distinct backscatter signatures. As a result, we employed a temporal analysis approach, or the seasonal wetland classification method, to generate an annual wetland map focusing on the backscatter coefficient from both time periods to leverage the seasonal changes in land surface characteristics.

**Methods of mapping of wetland indicators.** Among the three primary wetland classification systems—Ramsar International Wetland classification, the Cowardin et al. system, and the hydrogeomorphic (HGM) classification [32,33] —we adapted the HGM system for this study. This approach focuses on three key factors: landscape position, major water supply, and hydrodynamics [33], which were essential for mapping and monitoring changes in the lacustrine fringe wetlands of Lake Tana. We utilized advanced technology by integrating multi-temporal and multi-source data for the classification, delineation, and mapping of the lacustrine wetlands at Lake Tana. This involved employing a range of automated, semi-automated, and manual techniques. For the automated approaches, key inputs included major soil types (HS), DEM and their derivatives, LB, DEM/SRTM data, and SAR. In addition to these automated methods, we conducted on-screen digitization of the enhanced imagery, manual delineation based on field data, and the delineation, merging, and naming of wetland parcels in accordance with political boundaries. In this study, Arc GIS software version 10.7.1, AHP ArcGIS Toolbox, PCI Geomatica Banff (Focus Banff Edition, SP2, 2020-07-29), and ArcGIS Pro 3.4 were used. The general workflow of mapping shore area wetlands presented in the Fig 2.

*Topographic position.* Lacustrine fringe wetlands within the Lake Tana landscapes are created and sustained by lacustrine processes, primarily receiving water from the lake itself. Consequently, TP, particularly elevation, plays a crucial role in the formation of these wetlands. In this study, TP and LB are considered as an important indicator for mapping wetland areas along the shore. Bathymetric maps of wetlands serve many purposes, including defining legal boundaries, estimating water storage capacity and hydroperiod (depth and timing of flooding), as well as aiding in wetland design, restoration, and land use planning [34].

Secondary data from the bathymetry of Lake Tana during 2017 [35] was utilized to delineate the boundaries of the wetlands from the lake-ward side. In contrast, the average maximum lake level and elevation derivatives from the DEM were employed to identify the boundaries of these wetlands from the landward side. During the pre-field visit of 2021, the maximum distance of the extended shore area wetland from the Lake Tana shoreline during maximum water level is about 3-kilometer and has 1790 meters above sea level elevation. Thus, based on the maximum distance of the extended shore area wetland, we determined our study area to be a 3 km buffer zone around the lake. Besides, the elevation meter above

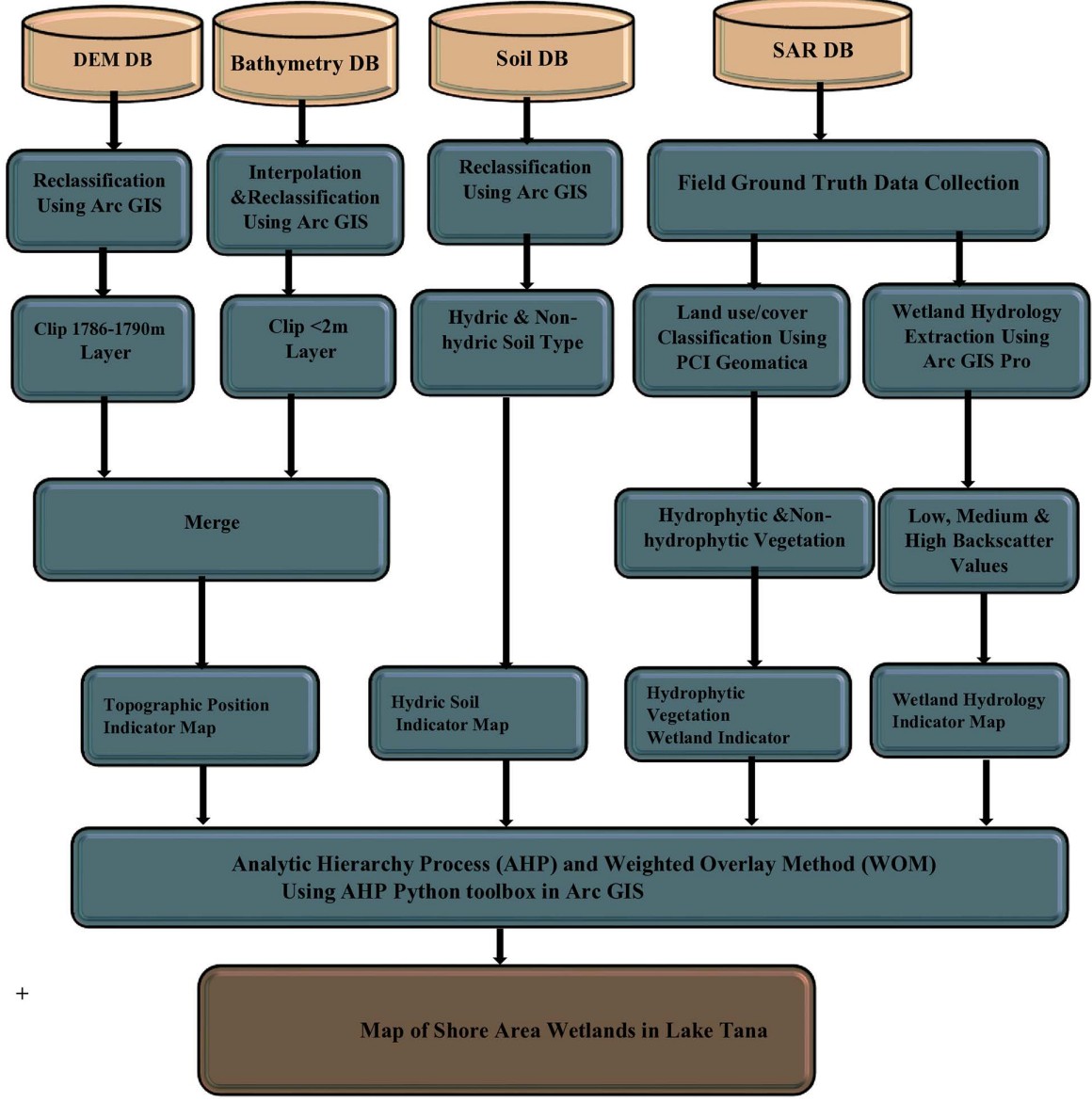

**Fig 2. General workflow of mapping shore area wetlands.**

sea level up to 1790 is considered as potential shore area wetland around Lake Tana, where wetlands receive water from an adjacent large freshwater lake, Tana.

The process of generating TP mapping involved several key steps (Fig 3): (a) reclassifying elevation maps into five categories within a 3 km buffer zone around the lake; (b) identifying critical elevation values for Lake Tana and its surroundings, specifically up to 1790 meters within the 3 km buffer, to delineate the boundaries of shoreline wetlands from the landward side; (c) reclassifying the lake bathymetry map into eight categories; (d) utilizing the first category (0–2 meters depth) from the bathymetry map to define the boundaries of shoreline wetlands from the lake-ward side (S5 Table); and (e) merging two layers using Data Management Tool of General: Merge—the elevation map for the range of 1786–1790 meters and the bathymetry map for depths of 0–2 meters—to create the TP wetland indicator map.

**Fig 3. Topographic position mapping processes.** (a) the critical elevation values of Lake Tana and the surrounding areas up to 1790 meters within the 3 km buffer of Lake Tana, which were used to extract the boundaries of the lacustrine wetlands from the terrestrial side; (b) the first class (0–2 meter depth) was used to extract the boundaries of the lacustrine wetlands from the lake-ward side. Then, the two layers (a and b) were merged to produce the TP wetland indicator map. Lake Tana Bathymetry remapped from [35] under a CC BY license, with permission from [John Wiley & Sons, Ltd.], original copyright [© 2021}.

*Hydric soils.* HSs are characterized by saturation, ponding, or flooding for a duration sufficient during the growing season to foster the development of anaerobic conditions in the upper layers [36].These conditions create an ideal environment for the growth and reproduction of HV. In contrast, non-hydric soils are generally less conducive to the formation of wetlands. However, it is important to note that the absence of HS does not necessarily indicate that wetlands are absent from the area [36].Thus, to identify areas with a higher likelihood of wetland presence in this study, the HS indicator was utilized alongside indicators of HV and TP.

Thus, to delineate HSs accurately, it was essential to employ national or regional field indicators of HSs or conduct a comprehensive soil survey. However, conducting detailed soil surveys for wetland mapping in this research project proved prohibitively expensive. As a result, we adopted a digital format of the Tana sub-basin soil survey for this study [37].

Three commonly recognized morphological traits can be utilized to differentiate between hydrophilic and non-hydrophilic soils: the presence of organic matter, gleying, and mottling or redoximorphic structures. By considering these established morphological characteristics, the major soil types surrounding Lake Tana within a 3 km buffer (Fig 4a) have been reclassified into a principal category labeled hydric, which includes Vertisols, Fluvisols, Gleysols, and Water (S6 Table). The remaining areas, comprising urban built-up zones, have been classified as non-hydric soils. Furthermore, these HSs are subdivided into water bodies, permanently hydric soils (Gleysols), and seasonally hydric soils (Vertisols and Fluvisols) (see the result section Fig 7b).

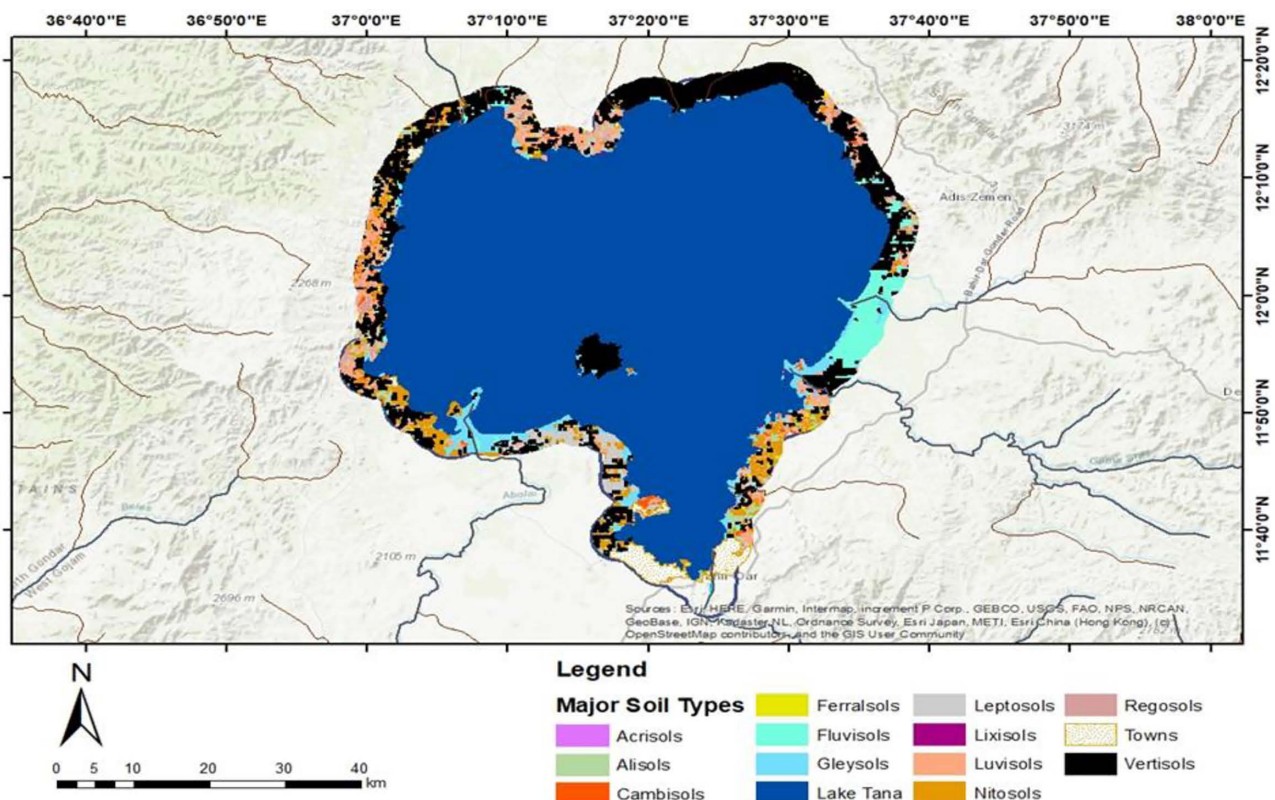

**Fig 4. Hydric soil mapping processes.** *Major soil types around Lake Tana up to a 3 km buffer extracted from a soil survey done by [37]; and major soil types reclassified into water bodies, permanently hydric soils (Gleysols), and seasonally hydric soils (Vertisols and Fluvisols) presented at result section(Fig 7b).*

***Mapping hydrophytic vegetation using SAR data.*** Wetlands are characterized by their distinctive HV [38]. To enhance the efforts of wetland delineation and mapping in the study area, a survey of HV was conducted and integrated into the mapping process. By utilizing data collected across multiple seasons, we can improve the differentiation of wetland types through an analysis of seasonal variations in phenology that reflect changes in plant structure. This study specifically utilized wetland vegetation data gathered in May 2021 and October 2021.

In this study, HV was mapped as a wetland indicator using Sentinel-1A SAR data in conjunction with Object Analyst, an add-on package for PCI Geomatica Banff software. Sentinel-1A SAR was chosen for mapping HV in the shore area wetlands of Lake Tana for several reasons: (1) Sentinel-1A SAR is particularly effective when the performance of optical sensors is compromised by cloud cover and varying day/night conditions [18]; (2) The SAR signal's ability to penetrate through vegetation and soil provides additional information that is not accessible through optical remote sensing data [15,22].

Within Geomatica Focus, Object Analyst provides a comprehensive interface that encompasses processes for segmenting imagery, extracting features, building training sites, classifying data (including the creation of classification rules), reshaping, and assessing accuracy (Fig 5).

***Preprocessing.*** Among the preprocessing steps, mosaicking was performed using the Mosaic Tool. The purpose of mosaicking was to join overlapping images to form a single and uniform image. We created a quality mosaic image using the general workflow presented at the help of PCI Geomatica Banff, 2020 [39], under Mosaic Tool. (1) We selected the orthorectified images that we want. (2) We prepared the imagery for mosaicking by performing color balancing, generating cutlines, and applying image normalization. (3) We evaluated the preview image created after preparing the mosaic. (4) We adjusted manually the mosaic in Mosaic Tool. (5) We reevaluated the preview image created after preparing the mosaic. (6) Finally, we generated the full-resolution mosaic. Besides, mosaicking clipping was executed through Focus,

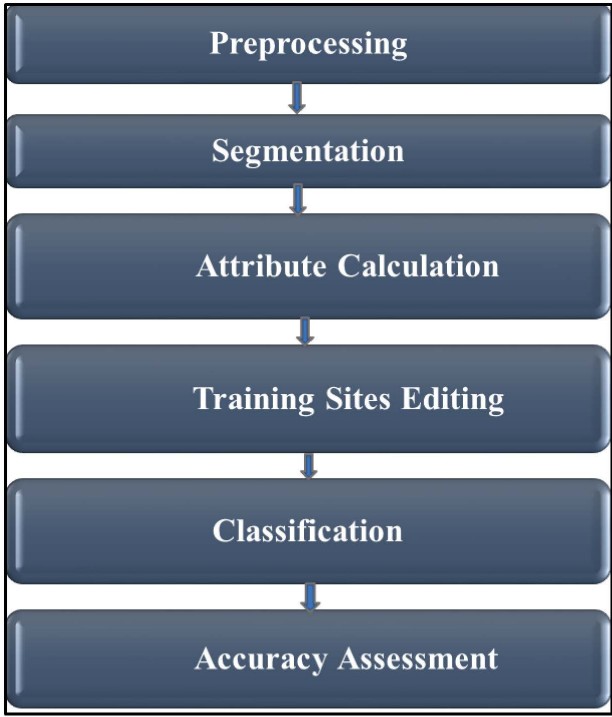

**Fig 5. Process flow for OBIA.** General workflow of hydrophytic vegetation mapping using object analyst within Geomatica Focus.

Tools: Clipping/Sub-setting. Using our mosaicked image as an input file, we clipped by our study area shape file. We defined the clip region by selecting a clip layer from the lists under Definition Method.

***Segmentation.*** Segmentation is regarded as a crucial and initial Object-Based Image Analysis (OBIA) phase [40,41].The image was segmented into statistically homogeneous areas or objects (segments) that are uniform within themselves and distinct from nearby segments, utilizing image segmentation techniques (Fig 5). Segmentation scale determines the size of objects; the compactness contributes to the fragmentation of landscape patches [39]. In this study, the segmentation optimum was defined by the following values: segmentation scale (25 pixels), shape (0.2) and compactness (0.5).

***Attribute calculation.*** Following segmentation, various attributes were calculated. For each segment (polygon defined by segmentation), attributes of statistical, geometric, and textural features were calculated. For each segment (polygon) created in the process of segmentation, four statistical (minimum, maximum, mean, and standard deviation) and five textural features (window size 11 × 11; mean, standard deviation, entropy, angular second moment, and contrast) were calculated [39].

***Training sites editing.*** Before we perform a classification or an accuracy assessment, we must have ground-truth data. In Object Analyst of PCI Geomatics, we could collect training samples for both the supervised classification and accuracy assessment in the same window of Training Sites Editing [39]. In the Training Sites Editing wizard, the sample type includes both training and accuracy assessment. Both training and accuracy polygons could be created by both manual edit and automatic import. In this study, we used manual or on-screen interpretation in the viewer and selected segments of various classes by selecting Create Polygon, Assign, Unassign, and Reset Class. Our ground truth shape files did not import automatically; rather, we used them for overlay purposes to create accurate training and accuracy sites. The data for accuracy should be of higher quality than the data used for training or classification [41,42].Thus, creating polygons selected for both training and accuracy assessment was carried out by applying GIS layer information created using the ground truth points gathered through field visits during May 2021 and October 2021 and SAS Planet imagery, resulting in these ground truth shape files overlaid with Sentinel-1A SAR images. Using these inputs, separate training and accuracy counts of polygons were created for both training and accuracy assessment using manual editing (S2 Table).

According to the criteria of wetland classification in the wetland convention and the current situation of land use/cover of shore area wetlands in Lake Tana surveyed during ground truth data collection, we classified the study area into nine land cover types: forest, built-up areas, invasive water hyacinth, cultivated land, hydrophytic vegetation, eucalyptus tree plantations, shrub land, and water bodies. A total of 4,728 training polygons representative of sites including forest (198), built-up areas (336), invasive water hyacinth (209), cultivated land (461), hydrophytic vegetation (603), eucalyptus tree plantations (64), shrub land (110), and water bodies (2747) were created. Similarly, a total of 3,176 accuracy polygons representative of sites including forest (87), built-up areas (190), invasive water hyacinth (143), cultivated land (443), hydrophytic vegetation (584), eucalyptus tree plantations (68), shrub land (100), and water bodies (1,561) were created. The number of polygons for training and accuracy by individual classes is shown in S2 Table.

***Hydrophytic and non-hydrophytic vegetation classification using SVM.*** We employed a temporal analysis approach, or the seasonal wetland classification method, to generate an annual hydrophytic wetland indicator map. Dry season, May 2021 image used because dry season images seem to have higher accuracy than the wet season images to successfully determine permanent shore area wetlands hydrophytic vegetation cover from the other land cover categories. However, seasonally flooded shore area wetlands hydrophytic vegetation classified as other covers mainly grasslands, and the map removed flooded wetlands and seasonal marshes. Therefore, we also considered the wet season, the October 2021 image that could include seasonal shore area wetlands hydrophytic cover. Finally, we updated the dry season map using Arc GIS software: Analysis Tools: Overlay: Update. We exported only the hydrophic vegetation cover categories from the wet season map to use as an update feature, while the dry season map was used as an input feature. The new updated annual hydrophic indicator map will be combined with the other wetland indicator to produce a shore area wetland map saved as an output feature class.

The classification function categorizes the segmented images into different land cover types based on the training data-set. Supervised classification was carried out using the Support Vector Machine technique [43,44].The advantage of this technique lies in its requirement for a minimal number of training samples [45]. The SVM classifier method was chosen in this study for the following reasons: (a) SVM is reported to perform better than Random Forest Trees (RFT) classifier when the training set is small or unbalanced. (b) When two classes are not discriminable linearly in a two-dimensional space, the kernel, a mathematical function used by the SVM classifier could separable in a higher-dimensional space (hyperplanes). From four basic kernels, the radial basis function (RBF) kernel provides the best results. With SVM, the objective is to find the optimal separating hyperplane (decision surface, boundaries) by maximizing the margin between classes, which is achieved by analyzing the training samples located at the edge of the potential class. Support Vector Machines (SVMs) are particularly effective for nonlinear classification challenges, which proves beneficial when extracting feature vectors from completely polarimetric SAR data. In this context, the radial basis function kernel was employed for classification, utilizing a high probability threshold of 0.9 and a penalty parameter of 100 [43].

***Classification accuracy assessments.*** Accuracy assessment is a measure of the agreement between a presumed standard that is considered accurate and image classification of unknown accuracy [39,41].Following the classification process, the land cover classification results, an accuracy assessment was conducted [46].The construction of a confusion matrix or error matrix is considered to be the basis of quantitative analysis in remote sensing [47]. The confusion matrix compares classified with reference or ground truth data [48]. The classification quality was assessed based on various metrics, including user's accuracy, producer's accuracy, overall accuracy (OAA), and the Kappa index of agreement (KIA) [46].The class accuracy statistics presented in the result section at Table 1 while the confusion matrix for this study presented at S7 Table.

***Wetland hydrology mapping using SAR data.*** Water determines the existence, establishment, and maintenance of specific types of wetlands and their associated activities [38].The distinct backscattering mechanisms exhibited by flooded plants and water enable the delineation of water bodies from surrounding vegetation [49].The varying roughness between water-covered areas and other land-cover types serves as the foundation for detecting surface waters [50]. From Sentinel-1 SAR GRD, features with higher backscatter appear brighter (high backscattering value ranges in decibels (dB)), and water beneath the vegetation in wetland ecosystems where radar pulses are backscattered twice, "double bounce" phenomenon, either from the water's surface to the vegetation or vice versa, appears semi-darker (medium backscattering value ranges in dB), while those with lower backscatter (low backscattering value ranges in dB) appear darker. Water bodies present very dark features that contrast sharply with their surroundings due to their minimal roughness and consequently low backscatter, as illustrated in the figure below (Fig 6). Sentinel-1 SAR provides Single Look Complex (SLC) data and Ground Range Detected (GRD) data for downloading. The GRD data were used to extract radar-backscattering coefficients. The data preprocessing was mainly carried out by ArcGIS Pro; with pixel classification performed using the Deep Learning tool available in the Image Analyst toolbox within ArcGIS Pro.

**Table 1. Class accuracy statistics.**

| Class Name | Producer Accuracy (%) | User Accuracy (%) | Kappa Statistic |
| --- | --- | --- | --- |
| Hydrophytic vegetation | 69.69 | 59.77 | 0.51 |
| Forest | 58.62 | 45.13 | 0.44 |
| Built-up | 56.32 | 57.22 | 0.54 |
| Eucalyptus plantation | 0.00 | 0.00 | −0.02 |
| Waterbody | 94.11 | 99.12 | 0.98 |
| Cultivated land | 77.20 | 81.04 | 0.78 |
| Shrub-land | 45.00 | 29.03 | 0.27 |
| Water hyacinth | 54.55 | 58.21 | 0.56 |

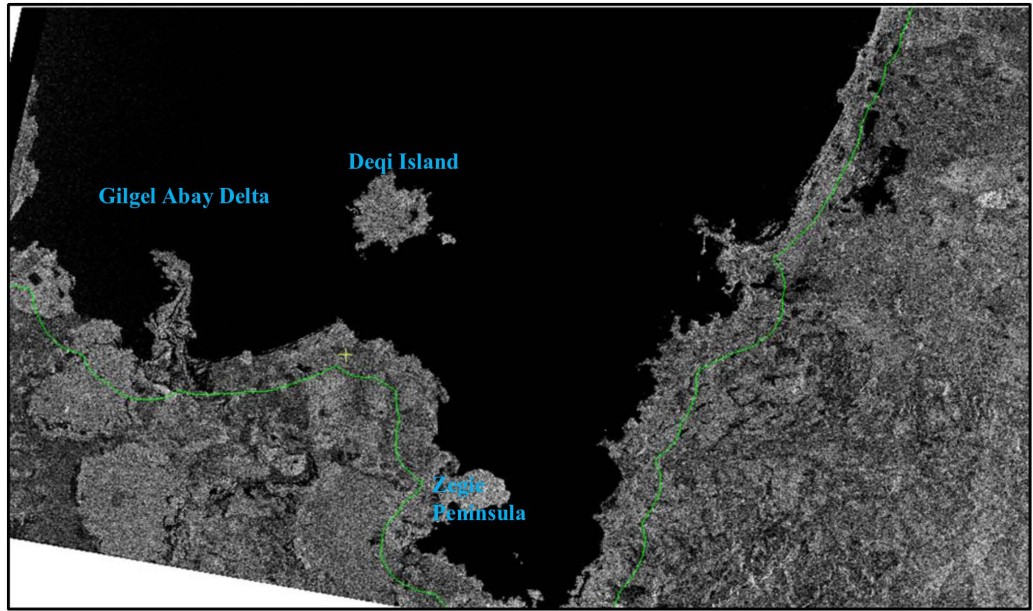

**Fig 6. The difference in backscattering between the Gulf of Lake Tana region covered by water and other types of land cover.** Water features appear as very dark features that contrast sharply with their surroundings land cover because they have very little roughness and, consequently, very low backscatter whereas the surrounding terrestrial parts including Zegie Peninsula, Deqi Island and Gilgel Abay Delta appear as white features because they have very high roughness and, consequently, very high backscatter and shore area wetlands around Lake Tana appear semi-dark because they have medium backscatter.

Permanent shore area wetlands, or those that are consistently inundated, are marked by a consistently low SAR backscatter in both the May and October SAR images. Our classification algorithm identifies these regions based on the stable low backscatter signature observed across both seasonal periods. Temporary/seasonal inundation of shore area wetlands category is characterized by a high change in backscatter between the two periods. We focus on areas that exhibit low backscatter in May, followed by a marked increase in backscatter in October. Finally, the annual wetland hydrology indicator map was produced by merging the two layers (temporary and permanently inundated layers produced from the dry and wet season images) using ArcGIS software: Data Management Tool of General: Merge.

**Integrating indicators for shore area wetlands mapping.** In efforts to facilitate the mapping of shore area wetlands at Lake Tana, four key wetland indicators were selected: WH, HV, HS, and TP. Each of these indicators was converted into raster maps utilizing the same coordinate system (UTM zone 37N) and a uniform pixel size of 30m x 30m.

The Analytic Hierarchy Process (AHP) and the weighted overlay method were employed to combine these indicators for the mapping (equation 1). This study applied the AHP Python toolbox integrated within ArcGIS 10.7, performing pairwise comparisons as outlined by Saaty in the Analytical Hierarchy Method [51]. Initially, the AHP toolbox was added to ArcGIS 10.7, followed by the incorporation of AHP script tools. The AHP script was used to create an empty AHP matrix, which we then filled in (S3 Table). The consistency index (CI) and consistency ratio (CR) were derived from the comparison matrix using the AHP script tool (S4 Table).

Subsequently, a comprehensive table was generated, detailing the weights assigned to each layer along with the CI, CR, and the average consistency index (RI). In this case, the CR (0.096) is below the threshold value of 1, indicating that the matrix is sufficiently consistent (S4 Table). The output raster could then be weighted according to importance and combined to produce a final output raster. The wetland indicators were assigned weights based on their contributions to wetland existence, with wetland hydrology recognized as the most critical variable among the three defining factors of

wetlands: WH, HS and HV. This is because suitable hydrologic conditions primarily determine the other characteristics. As a result, the greatest weight was assigned to WH, followed by HV.

$$Output = (WH) \times (0.50) + (HV) \times (0.29) + (HS) \times (0.14) + (TP) \times (0.07) \dots\dots\dots\dots\dots\dots\dots\dots\dots\dots\dots\dots\dots\dots\dots \quad (1)$$

## Results

### Area coverage of shore area wetlands according to individual indicators

**Topographic position.** In Fig 7a, the TP wetland indicator is illustrated, which was created by combining bathymetric data from depths of 0–2 meters with elevation maps ranging from 1786 to 1790 meters. This integration of data provides valuable insights into the ecological characteristics and spatial distribution of wetlands in this region. The topography position wetland indicator map covered about 55,363.53 ha.

**Hydric soil indicators.** The soils within a 3 km buffer zone around Lake Tana consist of 12 major types. These include Vertisols (34,924.07 ha), Luvisols (8,693.89 ha), Nitisols (8,725.15 ha), Leptosols (7,708.54 ha), Alisols (6,105.11 ha), Cambisols (4,524.73 ha), Regosols (957.38 ha), Fluvisols (9,760.32 ha), Ferralsols (1,016.11 ha), Gleysols (10,466.38 ha), Acrisols (156.19 ha), and Lixisols (56.75 ha) (Fig 7b).

Based on recognized morphological characteristics, these soils were reclassified into HSs (including Vertisols, Fluvisols, Gleysols, and water) and non-hydric soils (including urban and built-up areas). Hydric soils were further categorized into water bodies, permanently hydric soils (Gleysols), and seasonally hydric soils (Vertisols and Fluvisols). Together, the permanently and seasonally HSs within the 3 km buffer zone of Lake Tana cover approximately 55,151 ha (Fig 7b).

Fluvisols are predominantly located in depressions along major streams and on lower, gently sloping plains. These soils are subject to frequent flooding and are continually replenished with new sediments during annual floods. They are primarily found in the Rib, Gilgel Abay, Gumara, and Shini rivers, as well as the Fogera plains, and are associated with floodplain and lacustrine temporary wetlands (Fig 7b). Gleysols, which have hydromorphic properties within 50 cm of the surface, form under prolonged wet conditions that create reducing environments. These conditions lead to the transformation of ferric compounds into ferrous compounds, resulting in hydric characteristics. Gleysols in Lake Tana are closely associated with lacustrine permanent wetlands and water depths of less than two meters. However, not all areas with hydric soils, including Vertisols, Fluvisols, and Gleysols, meet the criteria for classification as shore area wetlands. Some lack the hydrology typical of wetlands or do not support hydrophytic vegetation, which are essential characteristics of wetlands.

**Hydrophytic vegetation indicators.** Fig 7c illustrates eight major land use/cover classes around Lake Tana, derived from Sentinel-1A SAR input data and analyzed using the Object Analyst add-on within PCI Geomatica Banff software. The resulting map highlights the effectiveness of Object Analysts in identifying diverse land use/cover types within a 3 km buffer zone from the lake. These classifications were restructured into two primary categories: hydrophytic vegetation (including hydrophytic vegetation and invasive water hyacinth) and non-hydrophytic vegetation (comprising forests, built-up areas, cultivated land, shrub-land, Eucalyptus plantations, and water bodies).

The HV of shore area wetlands is widespread, except in the northern and northwestern parts of the lake (Fig 7c) and (S8 Table).Within the 3 km buffer, hydrophytic vegetation, including invasive water hyacinth, accounts for approximately 74,771.86 ha. Forests, covering 3,209.08 ha, are concentrated in areas such as Zegie, Kunzila, Woleta Petros, and Gorgora. Built-up areas, totaling 3,505.47 ha, are notable in Bahir Dar, Delgi, Kunzila, Sekelet, Sey-Deber, and Zegie.

The lake itself, along with cultivated land (18,939.38 ha), constitutes a significant portion of the area (S8 Table). Shrubland, covering 8,434.68 ha, is prevalent in kebeles such as Abrehajerha, Agid Kirigna, Debranta, Dengel-Ber, Gobay Mariam, Kabe, Mangie, Merafit, Mitireha, Qorata, Tana Mistily, and Wagetera. Eucalyptus plantations (30.28 ha) encroach on wetland habitats in several kebeles, including Delgi, Estimut, Gorgora, Lijoma, Nabega, Zenzelima, Sekelet, Shum Abo, Tsizamba, Wagetera, and Weramit (Fig 7c) and (S8 Table).

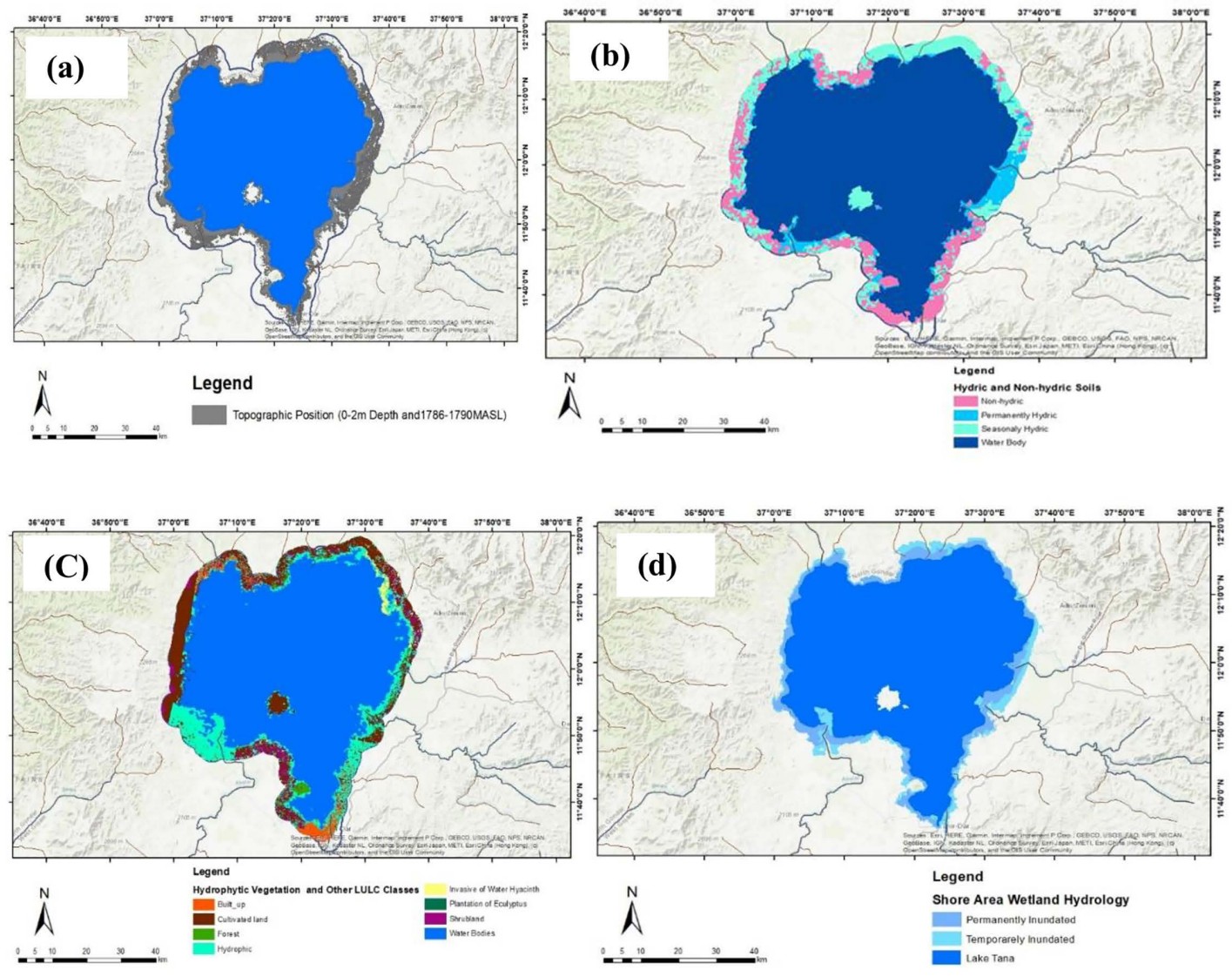

**Fig 7. Shore area wetland indicator maps.** (a) refers Topographic Position Wetland Indicator Map, (b) refers to Hydric Soil Wetland Indicator Map, (c) refers to Hydrophytic Vegetation Wetland Indicator Map; and (d) refers to Wetland Hydrology Indicator Map.

The overall classification accuracy and Kappa statistic were measured at 78.68% and 0.70, respectively. Additionally, we computed the Producer Accuracy (PA), User Accuracy (UA), and Kappa Statistic (KS) for each class as shown in Table 1. The results revealed that water bodies achieved the highest accuracies (PA=94.11%, UA=99.12%, KS=0.98), followed by cultivated land ((PA=77.20%, UA=81.04%, KS=0.780). HV showed moderate performance with PA at 69.69%, UA at 59.77%, and KS at 0.51. In contrast, shrub-land (PA=45.00%, UA=29.03%, KS=0.27) and Eucalyptus plantations (PA=0.00%, UA=0.00%, KS=−0.02) exhibited the lowest PA, UA, and KS values (Table 1).

***Wetland hydrology.*** Using Sentinel-1 SAR data, two WH categories were identified: (1) permanently inundated and (2) temporarily inundated areas. The hydrology wetland indicator map revealed that permanently inundated areas covered approximately 591,311.43 ha, while the inclusion of temporarily inundated areas increased the coverage to 607,052.48 ha (Fig 7d and Table 2).

**Table 2. Area coverage of shoreline wetlands according to individual indicators.**

| Wetland Indicator Type | Area(ha) |
| --- | --- |
| TP | 55,364 |
| HS | 55,151 |
| HV | 74,771 |
| WH | 607,052 |

### Shore area wetlands delineation using integrated wetland indicators

Shore area wetlands around Lake Tana extend approximately 3 km inland, covering 26,663.24 ha. (Fig 8). These wetlands are found in TPs below 1790 meters and are characterized by either permanent or temporary inundation. They include areas with hydromorphic soils and HV, which are adapted to periodic flooding and specific wet conditions.

Key wetland areas in the southern Gulf of Lake Tana include Aba-Garima, Selechen-Mariam, Avaji, Debo-Avanti, Giorgis-Alema, Shum-Abo, Gudguwad, and Gadero. Wetlands in the southwestern region encompass Dangel-Ber, Debranta, Estimut, the mouths of the Gilgel Abay and Infranz rivers, Kunzila, Lijoma-01, Lijoma-02, Sekelet, Sey-Deber, Wonjeta, and Yiganda-Zegie. In the southeast, wetlands such as Agid, Qirigna, Gileda, Bossit, the mouths of the Gumara and Rib rivers, Kabe, Merafit, Nabega, Qorata, Robit, Tana Mtsily, and Wagetera are prominent. Northern wetlands include Chemera, Chachna-Alwa, Mekonta-Ayibga, Fentaye-Narchacha, Delgi, Gorgora, the Dirma River Mouth, Tana-Woyna, Jerjer-Abanov, and Lemba-Arbaytu (Fig 8).

HV represents a critical structural element of the shore area wetland communities around Lake Tana. These wetlands can be classified into four groups based on the composition of HV:

1. **Southern Gulf of Lake Tana**: Dominated by reed macrophyte species.

2. **Northeastern Shore, Including the Fogera Floodplain**: Characterized by invasive water hyacinth macrophytes and a substrate of silt and sand sediments.

3. **Northern Dembia Floodplain**: Dominated by a combination of water hyacinth and *Echinochloa* spp., with silt as the primary sediment type.

4. **Northwestern Shore**: Predominantly composed of *Echinochloa* spp. and water hyacinth, with sand deposits as the dominant sediment type.

## Discussion

### Present mapping methods and findings in relations to previous studies

Our study provides concrete evidence that Sentinel-1A SAR radar datasets significantly enhance shoreline wetland mapping, aligning with previous studies utilizing synthetic aperture radar (SAR) for wetland mapping [16–18,50,52–54]. In this research, we successfully mapped HV and WH using publicly available Sentinel-1 SAR data processed with PCI Geomatica Banff software. Similarly, [50] employed multi-source Earth observation data, including RADARSAT-2 imagery processed through PCI Geomatica, for dynamic wetland mapping in the Great Lakes Basin. The use of SAR sensors, particularly Sentinel-1A, offers distinct advantages over optical sensors for shoreline wetland mapping. SAR can able to penetrate vegetation canopies and operate under all weather and illumination conditions, directly measuring the extent of flooded vegetation rather than inferring it from surface water extent [8]. Our findings align with prior studies indicating that multi-temporal SAR data enhances wetland classification [50,54,55].While SAR-based methods are effective, our results suggest that the integration of SAR with multi-source data, such as DEMs, can further improve classification accuracy, as highlighted by [53]. Despite these strengths, our classification of water hyacinth invasions yielded lower

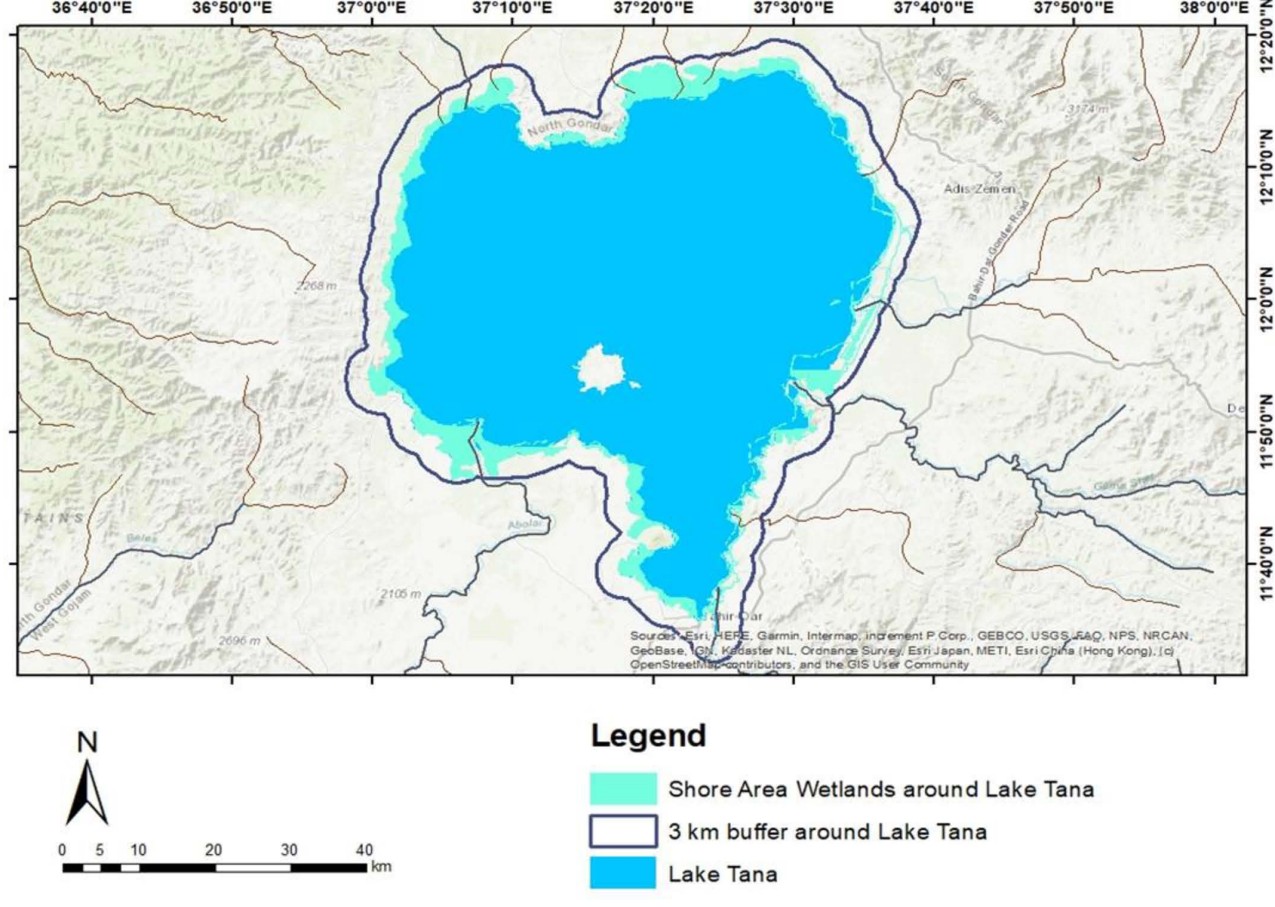

**Fig 8. Shore area wetland in Lake Tana.**

accuracy (PA = 54.55%, UA = 58.21%, KS = 0.56). OBIA methods combined with high-resolution data, as used by [50] for mapping Great Lakes coastal wetlands, have demonstrated superior accuracy [40]. OBIA approaches could potentially be enhanced by incorporating SAR data into the classification process.

This study emphasizes the utility of Sentinel-1A SAR and PCI Geomatica Banff for shoreline wetland mapping, particularly under conditions where optical sensors are limited by cloud cover or lighting [18]. SAR's ability to penetrate vegetation and soil provides valuable information unavailable to optical sensors [15,22]. However, sole reliance on SAR data may not always yield satisfactory accuracy. Integrating wetland indicators, such as hydric soils and topographic information derived from DEMs, can enhance mapping precision. Our findings contrast with [52], who reported poor performance of Sentinel-1A SAR for herbaceous wetland mapping in the Biebrza floodplain (Poland), citing coarse resolution and limitations in detecting small wetland features. They achieved an overall accuracy (OAA) of 65% and a Kappa index (KI) of 0.58 using multi-temporal SAR with VV/VH polarization. In comparison, our results demonstrated higher accuracy (OAA = 78.68%, KS = 0.70). Nonetheless, their findings highlighted the superior performance of TerraSAR-X/TanDEM-X images, especially in fully polarimetric mode.

Our classification results using Sentinel-1A SAR and PCI Geomatica showed good performance, with an overall accuracy of 78.68% and a Kappa coefficient of 0.70. These metrics are lower than those reported by [53] for wetland mapping in the Great Lakes (OA = 93.6%, K = 0.90), but higher than [50] 78% accuracy for targeted wetland classes. While our

study accurately identified hydrophytic classes, including cultivated land and water bodies, non-hydrophytic classifications remained less precise. In contrast, [53] achieved higher accuracy for non-wetland classifications (OA = 96.62%, K = 0.95) compared to wetland classifications (OA = 87%, K = 0.91). Our classifier employed a SVM method, which was also used by [52]. In contrast, [53] and [55] utilized the Random Forest (RF) classification algorithm, which has consistently been recognized as the most effective machine-learning approach for wetland mapping. RF classifiers have been particularly successful in categorizing wetlands into detailed classes, such as shallow water, marsh, swamp, fen, and bog, based on the Canadian Wetland Classification System [55].

The PCI Geomatica software used in this study supports various SAR sensors, including Sentinel-1, Radarsat-2, COSMO-SkyMed, TerraSAR-X, and others, offering extensive functionality for SAR data processing [50]. Geomatica's Object Analyst, an advanced module, provides an integrated workflow for segmenting imagery, extracting features, building training sites, classifying data, refining shapes, and conducting accuracy assessments [43,56–58]. However, its accessibility is constrained by licensing and copyright restrictions, limiting its application for open-source data analysis. In comparison, platforms like Google Earth Engine (GEE) offer free access to satellite datasets and enable scalable wetland mapping without the need for extensive local storage [53,55,58].Launched in 2014, the Sentinel-1 satellite is the most recent SAR mission operating at the C-band and it's freely available, which is another reason to explore the potential of this data for wetland InSAR applications [49].

## Limitations and future works

The proposed method of integrating Sentinel-1A SAR with multi-source, multi-temporal data shows promise for generating annual and seasonal wetland maps in the Lake Tana Biosphere Reserve. Combining SAR data with spatial wetland indicators, such as hydric soils and topographic position improves classification accuracy and enables more precise monitoring of shoreline wetland dynamics.

Although the study demonstrated that four wetland indicators (TP, HS, HV and WH) effectively map the spatial extent of shoreline wetlands in Lake Tana, northwestern Ethiopia, there are some limitations and remaining issues that might be addressed in future work. Factors that limit mapping of shore area wetlands in Lake Tana Biosphere Reserve were: (a) this study limited on lacustrine fringe wetland type's spatial scale and temporal scale (absence of change detection). From four multiple wetland indicators, hydric soil, hydrophytic vegetation, and wetland hydrology could be applied for all types of wetlands. However, the approach we applied to extract the topographic position indicator is restricted to lacustrine fringe and tidal fringe wetland types. Spatiotemporal change detection is another application of SAR wetland monitoring. The advent of high spatial resolution remote sensing imagery further supports opportunities to apply change detection with object-based change detection (OBCD). OBCD was not included in this study. This wetland change analysis would provide crucial information about drivers of lacustrine wetland changes in Lake Tana and, consequently, would improve better wetland management in the lacustrine wetland changes in Lake Tana. Therefore, changes in lacustrine wetlands in Lake Tana need to be detected using Sentinel-1 SAR data and Geomatica Banff software packages in future research work. (b) The total amount of carbon stored by each wetland could be estimated by multiplying the total area of the wetland with the value of carbon stock per hectare from the literature. Similarly, carbon emission could be estimated by multiplying the total area of the wetland with its corresponding emission/removal factor from the literature. However, this study focused on mapping area coverage and carbon storage as well as carbon emission were not included in this research work. Therefore, using the total area of the lacustrine wetlands in Lake Tana estimated by this study, the carbon storage and emission need to be estimated for two-wetland degradation states (pristine and drained conditions) in future research work. (c) This study depend on secondary data of LB and major soil type, which reveal mapping error. These factors need to be considered in future research. (d) SAR data processing in this study is supported by PCI Geomatica software, whose accessibility is constrained by licensing and copyright restrictions. Besides licensing and copyright restrictions, processing SAR data needs massive computation, which lowers the processing speed using a personal computer. Future research work needs

to be done to process SAR data using Google Earth Engine integrated with a large number of free remote sensing images and providing remote computation.

## Conclusions

This study demonstrated that four wetland indicators (TP, HS, HV and WH) effectively map the spatial extent of shoreline wetlands in Lake Tana, northwestern Ethiopia. The success of this approach is largely attributed to the "double bounce" phenomenon, where radar pulses are backscattered twice, enhancing the detection of wetland features. From an overall classification accuracy, we could conclude that hydrophytic classes including water bodies and cultivated-land were identified more accurately than non-hydrophytic classes.

The implications of these findings are profound for future research. The finding of this study suggests that SAR data alone could not produce satisfactory wetland classification accuracy. We minimize such limitations by integrating spatial geo-information indicators, such as TP and HSs derived from ArcGIS, with data captured by Sentinel-1A SAR, which focuses on hydrophytic vegetation and wetland hydrology, processed through PCI Geomatica Banff software. Mapping wetlands using multiple wetland indicator approach ensures a more accurate and comprehensive representation of shoreline wetlands. For future mapping, detecting changes and restoring shore area wetlands in Lake Tana, the study result provides important information on which wetland indicators (topographic position, hydric soil, hydrophytic vegetation and wetland hydrology) are highly significant indicators for the existence and restoration of wetland resources. The consistency ratio (CR) value of this study confirms that wetland hydrology could be recognized as the most critical variable among the three defining factors of wetlands. Because neither the characteristic substrates nor the typical biota of wetlands can emerge without appropriate hydrologic regimes. This implies that during mapping and restoring wetlands, priority needs to be given for wetland hydrology followed by hydrophytic vegetation.

The implications of these findings are also profound for future specific practical management recommendations. The results of this study provide a valuable baseline for future practical guidance for the sustainable management and conservation of shoreline wetland resources in the Lake Tana Biosphere Reserve. For managing and restoring shore area wetlands and making policy, the following measures are suggested: Shore area wetlands need to be delineated, and their buffer zone needs to be established using as a reference the results of this study.

## Supporting information

**S1 Table. Summary of data types and sources used in the study.**
(DOCX)

**S2 Table. Description of training and accuracy counts for this study.**
(DOCX)

**S3 Table. AHP matrix produced by AHP (step1) script and filled by the user.** *HS refers to Hydric Soil; HV refers to Hydrophytic Vegetation; WH refers to Wetland Hydrology; and TP refers to Topographic Position.*
(DOCX)

**S4 Table. AHP matrix produced by AHP (step2) script that generate weight, consistency index (CI), the average consistency index (RI), consistency ratio (CR) and Notes.**
(DOCX)

**S5 Table. Results of Lake Tana's bathymetry were interpolated using the Inverse Distance Weighting (IDW) method.**
(DOCX)

**S6 Table. Major soil types, hydric nature, sub-classes and their area coverage in hectare.**
(DOCX)

**S7 Table. Confusion matrix.**
(DOCX)

**S8 Table. Major land use/cover and their area coverage in hectare.**
(DOCX)

## Acknowledgments

We extend our sincere thanks to European Space Agency for providing access to Sentinel-1 Data product that were instrumental in advancing this research. The authors wish to thank the anonymous reviewers for their valuable comments and recommendations on the original version of this study.

## Author contributions

**Conceptualization:** Yirga Kebede Wondim, Ayalew Wondie Melese.

**Data curation:** Yirga Kebede Wondim.

**Formal analysis:** Yirga Kebede Wondim.

**Funding acquisition:** Ayalew Wondie Melese.

**Investigation:** Yirga Kebede Wondim, Ayalew Wondie Melese.

**Methodology:** Ayalew Wondie Melese.

**Project administration:** Ayalew Wondie Melese.

**Resources:** Ayalew Wondie Melese, Workiyie Worie Assefa.

**Software:** Yirga Kebede Wondim.

**Supervision:** Ayalew Wondie Melese, Workiyie Worie Assefa.

**Validation:** Yirga Kebede Wondim, Ayalew Wondie Melese, Workiyie Worie Assefa.

**Visualization:** Yirga Kebede Wondim.

**Writing – original draft:** Yirga Kebede Wondim.

**Writing – review & editing:** Ayalew Wondie Melese, Workiyie Worie Assefa.

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
