## [Decision Letter · Decision Letter 0]

7 Jun 2025

Dear Dr. Wondim,

The manuscript requires a complete revision and reorganisation, of both the text and the figures. Until the figures are completely revised and made leggible, and the text carefully revised for clarity, the manuscript will not be accepted for for review. Futhermore, there are clear shortcomings in the Methods and a more clear description of how the approach taken can be used in similar studies outside of the study area. Additional and more recent references should also be used to support the approaches used in the study.

We look forward to receiving your revised manuscript.

Kind regards,

Steven Arthur Loiselle

Academic Editor

PLOS ONE

Journal Requirements:

2. We note that your Data Availability Statement is currently as follows: All relevant data are within the manuscript and its supporting information files.

4. We note that Figures 1, 3, 4, 6, 7, 8, 9, in your submission contain [map/satellite] images which may be copyrighted. All PLOS content is published under the Creative Commons Attribution License (CC BY 4.0), which means that the manuscript, images, and Supporting Information files will be freely available online, and any third party is permitted to access, download, copy, distribute, and use these materials in any way, even commercially, with proper attribution. For these reasons, we cannot publish previously copyrighted maps or satellite images created using proprietary data, such as Google software (Google Maps, Street View, and Earth). For more information, see our copyright guidelines: http://journals.plos.org/plosone/s/licenses-and-copyright.

a. You may seek permission from the original copyright holder of Figures 1, 3, 4, 6, 7, 8, 9, to publish the content specifically under the CC BY 4.0 license.

5. We note you have included a table to which you do not refer in the text of your manuscript. Please ensure that you refer to Table 4 in your text; if accepted, production will need this reference to link the reader to the Table.

6. Please include a new copy of Table 1 and 2 in your manuscript; the current table is difficult to read. Please follow the link for more information: https://blogs.plos.org/plos/2019/06/looking-good-tips-for-creating-your-plos-figures-graphics/ .

Reviewers' comments:

Reviewer's Responses to Questions

**Comments to the Author**

1. Is the manuscript technically sound, and do the data support the conclusions?

Reviewer #1: Partly

Reviewer #2: Partly

2. Has the statistical analysis been performed appropriately and rigorously?

Reviewer #1: No

Reviewer #2: N/A

3. Have the authors made all data underlying the findings in their manuscript fully available?

Reviewer #1: No

Reviewer #2: No

4. Is the manuscript presented in an intelligible fashion and written in standard English?

Reviewer #1: Yes

Reviewer #2: No

Reviewer #1: The manuscript attempts to address an important challenge in delineating shoreline wetlands by integrating Sentinel-1 SAR data and multi-source indicators (topography, hydric soil, hydrophytic vegetation, and hydrology) to map Lake Tana’s wetlands. While the topic is relevant and the multi-indicator approach is logical, the manuscript suffers from methodological ambiguities and the current quality of the figures significantly detracts from the readability.

Some parts of the methodology are unclear and lack sound reasoning.

Lines 142-145. The choice of a 3 km buffer zone and a critical elevation of 1790 m requires detailed explanation.

Mapping Hydrophytic Vegetation Using SAR Data. In this section, the number and date of Sentinel-1 images used are unknown. The construction of training datasets is unclear. How to generate training site polygons from ground truth points? How to determine the classification categories? Since the discussion section mentioned that the random forest is considered the most effective machine-learning approach for wetland mapping, why is only SVM used here?

Wetland Hydrology Mapping Using SAR Data. Since the hydrophytic vegetation mapping section already includes the waterbody category, why do we need another map in this section to extract waterbodies? Again, it is unclear how to use Sentinel-1 images in this section.

Current wetland mapping methods are well-established. A dedicated section comparing this approach with prior methods would strengthen the case for novelty.

Although the manuscript is well-structured, the figures are terrible and affect readability. The key figure showing hydrophytic vegetation mapping (Figure 5) is even a combination of screenshots of the Geomatica Focus interface.

Figure 1. The color of the basemap is close to the color of the legend, which interferes with the reading of key information (Figs 1).

Figure 2. The demonstration of the proposed workflow should be improved. The arrows are inconsistent, and some text is obscured. Why is there a “Wetland InSAR” when the mapping method in this manuscript only uses SAR backscattering intensity?

Figure 3. The scale and extent of the subplots are inconsistent, making it difficult to compare their differences. In addition, it is hard to understand the overlap of the two layers (1786-90 Elevation meter and 0-2 meter Depth) in subplot e. Figure 7 has similar problems.

Tables 1 and 2 should not use screenshots.

Reviewer #2: Dear authors,

I have gone through the manuscript Mapping of Shore Area Wetlands in Lake Tana Biosphere Reserve, Ethiopia Using Sentinel-1A SAR and Multi-Source Data. The manuscript needs more attention of the author.

• The English language should be revised. There are many grammar errors and writing mistakes.

• It is recommended to fully spell out the in the first use of the term in a manuscript. Once the abbreviation has been introduced, it is appropriate to use it throughout the rest of the manuscript. So, readers will better understand the meaning of the abbreviation and its relevance to the topic at hand. Please revise all abbreviations

• Introduction section the introduction section should be supported by recent literature reviews that used remote sensing such as optical and radar data to detect water bodies: It is not the first study used this model in the research region. I think the authors could read them and cite these. please see

• Abou Samra RM, El-Barbary SM (2018) The use of remote sensing indices for detecting environmental changes: a case study of North Sinai, Egypt Spatial Information Research 26:679-689. https://doi.org/10.1007/s41324-018-0211-1

• Abou Samra R.M. (2022). Dynamics of human-induced lakes and their impact on land surface temperature in Toshka Depression, Western Desert, Egypt. Environmental Science and Pollution Research, 29, 20892-20905.

• Abou Samra RM, Ali R, Halder B, & Yaseen ZM (2024) Assessing the Catastrophic Environmental Impacts on Dam Breach Using Remote Sensing and Google Earth Engine. Water Resources Management:1-17.

• You should add a section about characteristics of your study area including causes of selection, and the meteorological conditions.

• The methods section should be well written. Please add section of sources and characteristics of your datasets. Please clarify your processing steps and software used to process your datasets.

• Please add the implication of the findings for future research

• This write-up should address the current limitations in this field of study.

**Do you want your identity to be public for this peer review?** For information about this choice, including consent withdrawal, please see our Privacy Policy

Reviewer #1: No

Reviewer #2: No

---

## [Author Response · Author response to Decision Letter 1]

22 Jul 2025

We submitted the revised manuscript entitled as, “Mapping of Shore Area Wetlands in Lake Tana Biosphere Reserve, Northwest Ethiopia Using Sentinel-1A SAR and Multi-Source Data”.

We considered carefully all comments suggested by the academic editor and reviewers. The following items when submitting with our revised manuscript: (a) Response to reviewer. A rebuttal letter that responds to each point raised by the academic editor and reviewers upload this letter as a separate file labeled 'Response to Reviewers; (b) A marked-up copy of our manuscript that highlights changes made to the original version also upload this as a separate file labeled 'Revised Manuscript with Track Changes’ indicated in purple color; and (c) an unmarked version of our revised paper without tracked changes. uploaded this as a separate file labeled 'Manuscript.'

Kind-regards,

Yirga Kebede Wondim(Lead Author)

---

## [Decision Letter · Decision Letter 1]

18 Aug 2025

Dear Dr. Wondim,

**These are specifically related to missing information regarding the selection of SAR images. Please see the comments from Review 1 below. **

We look forward to receiving your revised manuscript.

Kind regards,

Steven Arthur Loiselle

Academic Editor

PLOS ONE

**Journal Requirements:**

**Additional Editor Comments:**

The manuscript has been greatly improved and is nearly ready for publication. There are a couple methodological points that need to be clarified before publication.

Reviewers' comments:

Reviewer's Responses to Questions

**Comments to the Author**

Reviewer #1: (No Response)

Reviewer #2: All comments have been addressed

2. Is the manuscript technically sound, and do the data support the conclusions?

Reviewer #1: Yes

Reviewer #2: Yes

3. Has the statistical analysis been performed appropriately and rigorously?

Reviewer #1: Yes

Reviewer #2: Yes

4. Have the authors made all data underlying the findings in their manuscript fully available?

Reviewer #1: Yes

Reviewer #2: Yes

5. Is the manuscript presented in an intelligible fashion and written in standard English?

Reviewer #1: Yes

Reviewer #2: Yes

**Reviewer #1: ** The manuscript has greatly improved in terms of clarity and figures. Some minor comments are needed to be addressed before publication.

Mapping hydrophytic vegetation using SAR data. i. It is unclear how you use SAR images from two different periods (May and October). Do you composite them into a single image? If so, why do you select these two months? ii. How do you calculate vegetation indices, as you only mentioned SAR data?

Wetland hydrology mapping using SAR data. Are the same SAR images used as for HV mapping? If so, how are permanent and temporary inundation identified, and was done by comparing these two months?

Line 258. “water bodies” is duplicated.

**Reviewer #2: ** The authors addressed all the previous comments, and it is acceptable for publication from the scientific point of view.

**Do you want your identity to be public for this peer review?** For information about this choice, including consent withdrawal, please see our Privacy Policy

Reviewer #1: No

Reviewer #2: No

---

## [Author Response · Author response to Decision Letter 2]

3 Sep 2025

We submitted the revised manuscript entitled as, “Mapping of Shore Area Wetlands in Lake Tana Biosphere Reserve, Northwest Ethiopia Using Sentinel-1A SAR and Multi-Source Data”.

We considered carefully all comments suggested by the academic editor and reviewer-1. The following items when submitting with our revised manuscript: (a) Response to reviewer. A rebuttal letter that responds to each point raised by the academic editor and reviewers upload this letter as a separate file labeled 'Response to Reviewers';(b) A marked-up copy of our manuscript that highlights changes made to the original version also upload this as a separate file labeled 'Revised Manuscript with Track Changes’ indicated in purple color, and (c) an unmarked version of our revised paper without tracked changes upload this as a separate file labeled 'Manuscript'.

---

## [Editor Report · Decision Letter 2]

10 Sep 2025

Mapping of Shore Area Wetlands in Lake Tana Biosphere Reserve, Northwest Ethiopia Using Sentinel-1A SAR and Multi-Source Data

PONE-D-24-59965R2

Dear Dr. Wondim,

We’re pleased to inform you that your manuscript has been judged scientifically suitable for publication and will be formally accepted for publication once it meets all outstanding technical requirements.

Kind regards,

Steven Arthur Loiselle

Academic Editor

PLOS ONE
---

## [Editor Report · Acceptance letter]

PONE-D-24-59965R2

PLOS ONE

Dear Dr. Wondim,

I'm pleased to inform you that your manuscript has been deemed suitable for publication in PLOS ONE. Congratulations! Your manuscript is now being handed over to our production team.

Kind regards,

on behalf of

Dr. Steven Arthur Loiselle

Academic Editor

PLOS ONE